# Demystifying Black-box Models with Symbolic Metamodels

**Ahmed M. Alaa**
ECE Department
UCLA
ahmedmalaa@ucla.edu

**Mihaela van der Schaar**
UCLA, University of Cambridge, and
Alan Turing Institute
{mv472@cam.ac.uk,mihaela@ee.ucla.edu}

## Abstract

Understanding the predictions of a machine learning model can be as crucial as the model's accuracy in many application domains. However, the *black-box* nature of most highly-accurate (complex) models is a major hindrance to their interpretability. To address this issue, we introduce the *symbolic metamodeling* framework — a general methodology for interpreting predictions by converting "black-box" models into "white-box" functions that are understandable to human subjects. A symbolic metamodel is a *model of a model*, i.e., a surrogate model of a trained (machine learning) model expressed through a succinct symbolic expression that comprises familiar mathematical functions and can be subjected to symbolic manipulation. We parameterize metamodels using Meijer $G$-functions — a class of complex-valued contour integrals that depend on real-valued parameters, and whose solutions reduce to familiar algebraic, analytic and closed-form functions for different parameter settings. This parameterization enables efficient optimization of metamodels via gradient descent, and allows discovering the functional forms learned by a model with minimal a priori assumptions. We show that symbolic metamodeling provides a generalized framework for model interpretation — many common forms of model explanation can be analytically derived from a symbolic metamodel.

## 1 Introduction

The ability to interpret the predictions of a machine learning model brings about user trust and supports understanding of the underlying processes being modeled. In many application domains, such as the medical and legislative domains [1–3], model interpretability can be a crucial requirement for the deployment of machine learning, since a model's predictions would inform critical decision-making. Model explanations can also be central in other domains, such as social and natural sciences [4, 5], where the primary utility of a model is to help understand an underlying phenomenon, rather than merely making predictions about it. Unfortunately, most state-of-the-art models — such as ensemble models, kernel methods, and neural networks — are perceived as being complex "black-boxes", the predictions of which are too hard to be interpreted by human subjects [1, 6–16].

**Symbolic metamodeling.** In this paper, we approach the problem of model interpretation by introducing the *symbolic metamodeling* framework for expressing black-box models in terms of transparent mathematical equations that can be easily understood and analyzed by human subjects (Section 2). The proposed metamodeling procedure takes as an input a (trained) model — represented by a black-box function $f(\mathbf{x})$ that maps a feature $\mathbf{x}$ to a prediction $y$ — and retrieves a *symbolic metamodel* $g(\mathbf{x})$, which is meant to be an interpretable mathematical abstraction of $f(\mathbf{x})$. The metamodel $g(\mathbf{x})$ is a tractable symbolic expression comprising a finite number of familiar functions (e.g., polynomial, analytic, algebraic, or closed-form expressions) that are combined via elementary arithmetic operations (i.e., addition and multiplication), which makes it easily understood by inspection, and can be

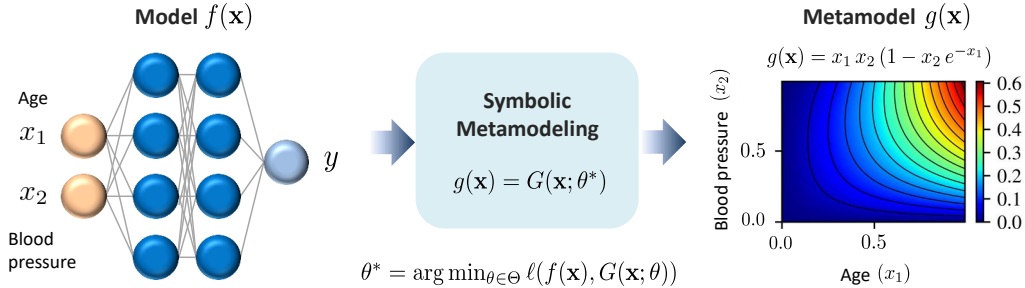

Figure 1: **Pictorial depiction of the symbolic metamodeling framework.** Here, the model $f(\mathbf{x})$ is a deep neural network (left), and the metamodel $g(\mathbf{x})$ is a closed-form expression $x_1\, x_2\, (1 - x_2\, \exp(-x_1))$ (right).

analytically manipulated via symbolic computation engines such as `Mathematica` [17], `Wolfram alpha` [18], or `Sympy` [19]. Our approach is appropriate for models with small to moderate number of features, where the physical interpretation of these features are of primary interest.

A high-level illustration of the proposed metamodeling approach is shown in Figure 1. In this Figure, we consider an example of using a neural network to predict the risk of cardiovascular disease based on a (normalized) feature vector $\mathbf{x} = (x_1, x_2)$, where $x_1$ is a person's age and $x_2$ is their blood pressure. For a clinician using this model in their daily practice or in the context of an epidemiological study, the model $f(\mathbf{x})$ is completely obscure — it is hard to explain or draw insights into the model's predictions, even with a background knowledge of neural networks. On the other hand, the metamodel $g(\mathbf{x}) = x_1\, x_2\, (1 - x_2\, \exp(-x_1))$ is a fully transparent abstraction of the neural network model, from which one can derive explanations for the model's predictions through simple analytic manipulation, without the need to know anything about the model structure and its inner workings[1]. Interestingly enough, having such an explicit (simulatable) equation for predicting risks is already required by various clinical guidelines to ensure the transparency of prognostic models [21].

**Metamodeling with Meijer $G$-functions.** In order to find the symbolic metamodel $g(\mathbf{x})$ that best approximates the original model $f(\mathbf{x})$, we need to search a space of mathematical expressions and find the expression that minimizes a "metamodeling loss" $\ell(g(\mathbf{x}), f(\mathbf{x}))$. But how can we construct a space of symbolic expressions without predetermining its functional from? In other words, how do we know that the metamodel $g(\mathbf{x}) = x_1\, x_2\, (1 - x_2 \exp(-x_1))$ in Figure 1 takes on an exponential form and not, say, a trigonometric or a polynomial functional form?

To answer this question, we introduce a novel parameterized representation of symbolic expressions (Section 3), $G(\mathbf{x}; \theta)$, which reduces to most familiar functional forms — e.g., arithmetic, polynomial, algebraic, closed-form, and analytic expressions, in addition to special functions, such as Bessel functions and Hypergeometric functions — for different settings of a real-valued parameter $\theta$. The representation $G(\mathbf{x}; \theta)$ is based on Meijer $G$-functions [22–24], a class of contour integrals used in the mathematics community to find closed-form solutions for otherwise intractable integrals. The proposed Meijer $G$-function parameterization enables minimizing the metamodeling loss efficiently via gradient descent — this is a major departure from existing approaches to *symbolic regression*, which use genetic programming to select among symbolic expressions that comprise a small number of predetermined functional forms [25–27].

**Symbolic metamodeling as a gateway to all explanations.** Existing methods for model interpretation focus on crafting explanation models that support only one "mode" of model interpretation. For instance, methods such as DeepLIFT [8] and LIME [16], can explain the predictions of a model in terms of the contributions of individual features to the prediction, but cannot tell us whether the model is nonlinear, or whether statistical interactions between features exist. Other methods such as GA$^2$M [9] and NIT [13], focus exclusively on uncovering the statistical interactions captured by the model, which may not be the most relevant mode of explanation in many application domains. Moreover, none of the existing methods can uncover the functional forms by which a model captures nonlinearities in the data — such type of interpretation is important in applications such as applied physics and material sciences, since researchers in these fields focus on distilling an analytic law that describes how the model fits experimental data [4, 5].

Our perspective on model interpretation departs from previous works in that, a symbolic metamodel $g(\mathbf{x})$ is not hardwired to provide any specific type of explanation, but is rather designed to provide a full mathematical description of the original model $f(\mathbf{x})$. In this sense, symbolic metamodeling should be understood as a tabula rasa upon which different forms of explanations can be derived — as we will show in Section 4, most forms of model explanation covered in previous literature can be arrived at through simple analytic manipulation of a symbolic metamodel.

## 2 Symbolic Metamodeling

Let $f : \mathcal{X} \to \mathcal{Y}$ be a machine learning model trained to predict a target outcome $y \in \mathcal{Y}$ on the basis of a $d$-dimensional feature instance $\mathbf{x} = (x_1, \ldots, x_d) \in \mathcal{X}$. We assume that $f(.)$ is a *black-box* model to which we only have query access, i.e., we can evaluate the model's output $y = f(\mathbf{x})$ for any given feature instance $\mathbf{x}$, but we do not have any knowledge of the model's internal structure. Without loss of generality, we assume that the feature space $\mathcal{X}$ is the unit hypercube, i.e., $\mathcal{X} = [0,1]^d$.

**The metamodeling problem.** A *symbolic metamodel* $g \in \mathcal{G}$ is a "model of the model" $f$ that approximates $f(\mathbf{x})$ for all $\mathbf{x} \in \mathcal{X}$, where $\mathcal{G}$ is a class of succinct mathematical expressions that are understandable to users and can be analytically manipulated. Typically, $\mathcal{G}$ would be set as the class of all arithmetic, polynomial, algebraic, closed-form, or analytic expressions. Choice of $\mathcal{G}$ will depend on the desired complexity of the metamodel, which in turn depends on the application domain. For instance, in experimental physics, special functions — such as Bessel functions — would be considered interpretable [4], and hence we can take $\mathcal{G}$ to be the set of all analytic functions. On the contrary, in medical applications, we might opt to restrict $\mathcal{G}$ to algebraic expressions. Given $\mathcal{G}$, the metamodeling problem consists in finding the function $g$ in $\mathcal{G}$ that bests approximates the model $f$.

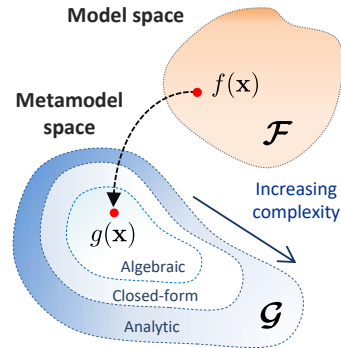

Figure 2: The metamodeling problem.

Figure 2 shows a pictorial depiction of the metamodeling problem as a mapping from the *modeling space* $\mathcal{F}$ — i.e., the function class that the model $f$ inhabits[2] — to the interpretable metamodeling space $\mathcal{G}$. Metamodling is only relevant when $\mathcal{F}$ spans functions that are considered uninterpretable to users. For models that are deemed interpretable, such as linear regression, $\mathcal{F}$ will already coincide with $\mathcal{G}$, because the linear model is already an algebraic expression (and a first-order polynomial). In this case, the best metamodel for $f$ is the model $f$ itself, i.e., $g = f$.

Formally, the metamodeling problem can be defined through the following optimization problem:

$$g^* = \arg\min_{g \in \mathcal{G}} \ell(g, f), \quad \ell(g, f) = \| f - g \|_2^2 = \int_{\mathcal{X}} (g(\mathbf{x}) - f(\mathbf{x}))^2 \, d\mathbf{x}, \tag{1}$$

where $\ell(.)$ is the *metamodeling loss*, which we set to be the mean squared error (MSE) between $f$ and $g$. In the following Section, we will focus on solving the optimization problem in (1).

## 3 Metamodeling via Meijer $G$-functions

In order to solve the optimization problem in (1), we need to induce some structure into the metamodeling space $\mathcal{G}$. This is obviously very challenging since $\mathcal{G}$ encompasses infinitely many possible mathematical expressions with very diverse functional forms. For instance, consider the exemplary metamodel in Figure 1, where $g(\mathbf{x}) = x_1 \, x_2 \, (1 - x_2 \exp(-x_1))$. If $\mathcal{G}$ is set to be the space of all closed-form expressions, then it would include all polynomial, hyperbolic, trigonometric, logarithmic functions, rational and irrational exponents, and any combination thereof [28, 29]. Expressions such as $g'(\mathbf{x}) = (x_1^2 + x_2^2)$ and $g''(\mathbf{x}) = \sin(x_1) \cdot \cos(x_2)$ are both valid metamodels, i.e., $g', g'' \in \mathcal{G}$, yet they each have functional forms that are very different from $g$. Thus, we need to parameterize $\mathcal{G}$ in such a way that it encodes all such functional forms, and enables an efficient solution to (1).

To this end, we envision a parameterized metamodel $g(\mathbf{x}) = G(\mathbf{x}; \theta)$, $\theta \in \Theta$, where $\Theta = \mathbb{R}^M$ is a parameter space that fully specifies the metamodeling space $\mathcal{G}$, i.e., $\mathcal{G} = \{G(.; \theta) : \theta \in \Theta\}$. Such parameterization should let $G(\mathbf{x}; \theta)$ reduce to different functions for different settings of $\theta$ — for the aforementioned example, we should have $G(\mathbf{x}; \theta') = (x_1^2 + x_2^2)$ and $G(\mathbf{x}; \theta'') = \sin(x_1) \cdot \cos(x_2)$ for some $\theta', \theta'' \in \Theta$. Given the parameterization $G(\mathbf{x}; \theta)$, the problem in (1) reduces to

$$g^*(\mathbf{x}) = G(\mathbf{x}; \theta^*), \text{ where } \theta^* = \arg\min_{\theta \in \Theta} \ell(G(\mathbf{x}; \theta), f(\mathbf{x})). \tag{2}$$

Thus, if we have a parameterized symbolic expression $G(\mathbf{x}; \theta)$, then the metamodeling problem boils down to a straightforward parameter optimization problem. We construct $G(\mathbf{x}; \theta)$ in Section 3.1.

### 3.1 Parameterizing symbolic metamodels with Meijer $G$-functions

We propose a parameterization of $G(\mathbf{x}; \theta)$ that includes two steps. The first step involves decomposing the metamodel $G(\mathbf{x}; \theta)$ into a combination of univariate functions. The second step involves modeling these univariate functions through a very general class of special functions that includes most known familiar functions as particular cases. Both steps are explained in detail in what follows.

**Step 1: Decomposing the metamodel.** We breakdown the *multivariate* function $g(\mathbf{x})$ into simpler, *univariate* functions. From the *Kolmogorov superposition theorem* [30], we know that every multivariate continuous function $g(\mathbf{x})$ can be written as a finite composition of univariate continuous functions and the addition operation as follows[3]:

$$g(\mathbf{x}) = g(x_1, \ldots, x_n) = \sum_{i=0}^{r} g_i^{out}\left(\sum_{j=1}^{d} g_{ij}^{in}(x_j)\right), \tag{3}$$

where $g_i^{in}$ and $g_{ij}^{out}$ are continuous univariate *basis functions*, and $r \in \mathbb{N}_+$. The exact decomposition in (3) always exists for $r = 2d$, and for some basis functions $g_i^{out} : \mathbb{R} \to \mathbb{R}$, and $g_{ij}^{in} : [0,1] \to \mathbb{R}$ [36]. When $r = 1$, (3) reduces to the generalized additive model [37]. While we proceed our analysis with the general formula in (3), in our practical implementation we set $r = 1$, $g^{out}$ as the identify function, and include extra functions $g_{ij}^{in}$ of the interactions $\{x_i x_j\}_{i,j}$ to account for the complexity of $g(\mathbf{x})$.

**Step 2: Meijer $G$-functions as basis functions.** Based on the decomposition in (3), we can now parameterize metamodels in terms of their univariate bases, i.e., $G(\mathbf{x}; \theta) = G(\mathbf{x}; \{g_i^{out}\}_i, \{g_{ij}^{in}\}_{i,j})$, where every selection of a different set of bases would lead to a different corresponding metamodel. However, in order to fully specify the parameterization $G(\mathbf{x}; \theta)$, we still need to parameterize the basis functions themselves in terms of real-valued parameters that we can practically optimize, while ensuring that the corresponding parameter space spans a wide range of symbolic expressions.

To fully specify $G(\mathbf{x}; \theta)$, we model the basis functions in (3) as instances of a Meijer $G$-function — a *univariate* special function given by the following line integral in the complex plane $s$ [22, 23]:

$$G_{p,q}^{m,n}\left(\begin{smallmatrix} a_1,\ldots,a_p \\ b_1,\ldots,b_q \end{smallmatrix} \middle| x\right) = \frac{1}{2\pi i} \int_{\mathcal{L}} \frac{\prod_{j=1}^{m} \Gamma(b_j - s) \prod_{j=1}^{n} \Gamma(1 - a_j + s)}{\prod_{j=m+1}^{q} \Gamma(1 - b_j + s) \prod_{j=n+1}^{p} \Gamma(a_j + s)} x^s \, ds, \tag{4}$$

where $\Gamma(.)$ is the Gamma function and $\mathcal{L}$ is the integration path in the complex plane. (In Appendix A, we provide conditions for the convergence of the integral in (4), and the detailed construction of the integration path $\mathcal{L}$.) The contour integral in (4) is known as Mellin-Barnes representation [24]. An instance of a Meijer $G$-function is specified by the real-valued parameters $\mathbf{a}_p = (a_1, \ldots, a_p)$, $\mathbf{b}_q = (b_1, \ldots, b_q)$, and indexes $n$ and $m$, which define the *poles* and *zeros* of the integrand in (4) on the complex plane[4]. In the rest of the paper, we refer to Meijer $G$-functions as $G$ functions for brevity.

For each setting of $\mathbf{a}_p$ and $\mathbf{b}_q$, the integrand in (4) is configured with different poles and zeros, and the resulting integral converges to a different function of $x$. A powerful feature of the $G$ function is that it encompasses most familiar functions as special cases [24] — for different settings of $\mathbf{a}_p$ and $\mathbf{b}_q$, it reduces to almost all known elementary, algebraic, analytic, closed-form and special functions.

Examples for special values of the poles and zeros for which the $G$ function reduces to familiar functions are shown in Table 1. (A more elaborate Table of equivalencies is provided in Appendix A.) Perturbing the poles and zeros around their values in Table 1 gives rise to variants of these functional forms, e.g., $x \log(x)$, $\sin(x)$, $x^2 e^{-x}$, etc. A detailed illustrative example for the different symbolic expressions that $G$ functions take on a 2D parameter space is provided in Appendix A. Tables of equivalence between $G$ functions and familiar functions can be found in [38], or computed using programs such as `Mathematica` [17] and `Sympy` [19].

| $G$-function | Equivalent form |
|---|---|
| $G_{3,1}^{0,1}\left(\begin{smallmatrix}2,2,2\\1\end{smallmatrix}\,\middle|\,x\right)$ | $x$ |
| $G_{0,1}^{1,0}\left(\begin{smallmatrix}-\\0\end{smallmatrix}\,\middle|\,x\right)$ | $e^{-x}$ |
| $G_{2,2}^{1,2}\left(\begin{smallmatrix}1,1\\1,0\end{smallmatrix}\,\middle|\,x\right)$ | $\log(1+x)$ |
| $G_{0,2}^{1,0}\left(\begin{smallmatrix}-\\0,\frac{1}{2}\end{smallmatrix}\,\middle|\,\frac{x^2}{4}\right)$ | $\frac{1}{\sqrt{\pi}}\cos(x)$ |
| $G_{2,2}^{1,2}\left(\begin{smallmatrix}\frac{1}{2},1\\\frac{1}{2},0\end{smallmatrix}\,\middle|\,x\right)$ | $2\arctan(x)$ |

Table 1: Representation of familiar elementary functions in terms of the $G$ function.

By using $G$ functions as univariate basis functions ($g_i^{in}$ and $g_{ij}^{out}$) for the decomposition in (3), we arrive at the following parameterization for $G(\mathbf{x};\theta)$:

$$G(\mathbf{x};\theta) = \sum_{i=0}^{r} G_{p,q}^{m,n}\left(\theta_i^{out} \,\middle|\, \sum_{j=1}^{d} G_{p,q}^{m,n}\left(\theta_{ij}^{in} \,\middle|\, x_j\right)\right), \tag{5}$$

where $\theta = (\theta^{out}, \theta^{in})$, $\theta^{out} = (\theta_0^{out}, \ldots, \theta_r^{out})$ and $\theta^{in} = \{(\theta_{i1}^{in}, \ldots, \theta_{id}^{in})\}_i$ are the $G$ function parameters. Here, we use $G_{p,q}^{m,n}(\theta \,|\, x) = G_{p,q}^{m,n}(\mathbf{a}_p, \mathbf{b}_q \,|\, x)$, $\theta = (\mathbf{a}_p, \mathbf{b}_q)$, as a shortened notation for the $G$ function for convenience. The indexes $(m, n, p, q, r)$ are viewed as hyperparameters of the metamodel.

**Symbolic metamodeling in action.** To demonstrate how the parameterization $G(\mathbf{x};\theta)$ in (5) captures symbolic expressions, we revisit the stylized example in Figure 1. Recall that in Figure 1, we had a neural network model with two features, $x_1$ and $x_2$, and a metamodel $g(\mathbf{x}) = x_1 x_2 (1 - x_2 e^{-x_1})$. In what follows, we show how the metamodel $g(\mathbf{x})$ can be arrived at from the parameterization $G(\mathbf{x};\theta)$.

Figure 3 shows a schematic illustration for the parameterization $G(\mathbf{x};\theta)$ in (5) — with $r = 2$ — put in the format of a "computation graph". Each box in this graph corresponds to one of the basis functions $\{g_i^{in}\}_i$ and $\{g_{ij}^{out}\}_{i,j}$, and inside each box, we show the corresponding instance of $G$ function that is needed to give rise to the symbolic expression $g(\mathbf{x}) = x_1 x_2 (1 - x_2 e^{-x_1})$. To tune the poles and zeros of each of the 6 $G$ functions in Figure 3 to the correct values, we need to solve the optimization problem in (2). In Section 3.2, we show that this can be done efficiently via gradient descent.

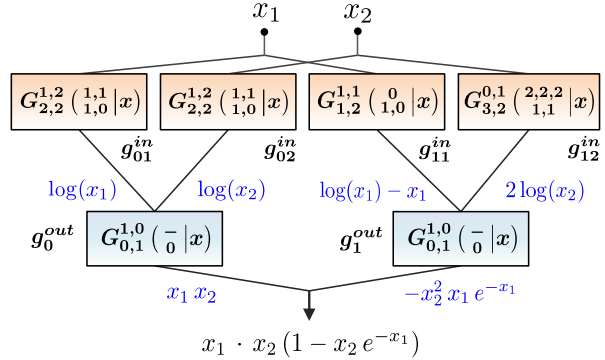

Figure 3: Schematic for the metamodel in Figure 1.

## 3.2 Optimizing symbolic metamodels via gradient descent

Another advantage of the parameterization in (5) is that the gradients of the $G$ function with respect to its parameters can be approximated in analytic form as follows [24]:

$$\frac{d}{da_k} G_{p,q}^{m,n}\left(\begin{smallmatrix}\mathbf{a}_p\\\mathbf{b}_q\end{smallmatrix}\,\middle|\,x\right) \approx x^{a_k-1} \cdot G_{p+1,q}^{m,n+1}\left(\begin{smallmatrix}-1,a_1-1,\ldots,a_n-1,a_{n+1}-1,\ldots,a_p-1\\b_1,\ldots,b_m,b_{m+1},\ldots,b_q\end{smallmatrix}\,\middle|\,x\right), \ 1 \le k \le p,$$

$$\frac{d}{db_k} G_{p,q}^{m,n}\left(\begin{smallmatrix}\mathbf{a}_p\\\mathbf{b}_q\end{smallmatrix}\,\middle|\,x\right) \approx x^{1-b_k} \cdot G_{p,q+1}^{m,n}\left(\begin{smallmatrix}a_1,\ldots,a_n,a_{n+1},\ldots,a_p\\b_1-1,\ldots,b_m-1,0,b_{m+1}-1,\ldots,b_q-1\end{smallmatrix}\,\middle|\,x\right) \ 1 \le k \le q. \tag{6}$$

From (6), we see that the approximate gradient of a $G$ function is also a $G$ function, and hence the optimization problem in (2) can be solved efficiently via standard gradient descent algorithms.

The solution to the metamodel optimization problem in (2) must be confined to a predefined space of mathematical expressions $\mathcal{G}$. In particular, we consider the following classes of expressions:

**Polynomial expressions** $\subset$ **Algebraic expressions** $\subset$ **Closed-form expressions** $\subset$ **Analytic expressions**,

where the different classes of mathematical expressions correspond to different levels of metamodel complexity, with polynomial metamodels being the least complex (See Figure 2).

Algorithm 1 summarizes all the steps involved in solving the metamodel optimization problem. The algorithm starts by drawing $n$ feature points uniformly at random from the feature space $[0,1]^d$ — these feature points are used to evaluate the predictions of both the model and the metamodel in order to estimate the metamodeling loss in (1). Gradient descent is then executed using the gradient estimates in (6) until convergence. (Any variant of gradient descent can be used.) We then check if every basis function in the resulting metamodel $g(\mathbf{x})$ lies in $\mathcal{G}$. If $g(\mathbf{x}) \notin \mathcal{G}$, we search for an approximate version of the metamodel $\tilde{g}(\mathbf{x}) \approx g(\mathbf{x})$, such that $\tilde{g}(\mathbf{x}) \in \mathcal{G}$.

---

**Algorithm 1** Symbolic Metamodeling

---

■ **Input:** Model $f(\mathbf{x})$, hyperparameters $(m, n, p, q, r)$

■ **Output:** Metamodel $g(\mathbf{x}) \in \mathcal{G}$

---

● $X_i \sim \text{Unif}([0,1]^d)$, $i = \{1, \ldots, n\}$.

● **Repeat until convergence:**

$$\theta^{k+1} := \theta^k - \gamma \nabla_\theta \sum_i \ell(G(X_i; \theta), f(X_i))\big|_{\theta=\theta_k}$$

● $g(\mathbf{x}) \leftarrow G(X_i; \theta^k)$

● **If** $g(\mathbf{x}) \notin \mathcal{G}$:

$\tilde{g}(\mathbf{x}) = G(\mathbf{x}; \bar{\theta})$, $G(\mathbf{x}; \bar{\theta}) \in \mathcal{G}$, $\|\bar{\theta} - \theta^k\| < \delta$, or
$\tilde{g}(\mathbf{x}) = \text{Chebyshev}(g(\mathbf{x}))$

---

The approximate metamodel $\tilde{g}(\mathbf{x}) \in \mathcal{G}$ is obtained by randomly perturbing the optimized parameter $\theta$ (within a Euclidean ball of radius $\delta$) and searching for a valid $\tilde{g}(\mathbf{x}) \in \mathcal{G}$. If no solution is found, we resort to a Chebyshev polynomial approximation of $g(\mathbf{x})$ — we can also use the Taylor or Padé approximations — since polynomials are valid algebraic, closed-form and analytic expressions.

## 4 Related Work: Symbolic Metamodels as Gateways to Interpretation

The strand of literature most relevant to our work is the work on *symbolic regression* [25–27]. This is a regression model that searches a space of mathematical expressions using *genetic programming*. The main difference between this method and ours is that symbolic regression requires predefining the functional forms to be searched over, hence the number of its parameters increases with the number of functions that it can fit. On the contrary, our Meijer $G$-function parameterization enables recovering infinitely many functional forms through a fixed-dimensional parameter space, and allows optimizing metamodels via gradient descent. We compare our method with symbolic regression in Section 5.

**Symbolic metamodeling as a unifying framework for interpretation.** We now demonstrate how symbolic metamodeling can serve as a *gateway* to the different forms of model explanation covrered in the literature. To vivify this view, we go through common types of model explanation, and show that given a metamodel $g(\mathbf{x})$ we can recover these explanations via analytic manipulation of $g(\mathbf{x})$.

The most common form of model explanation involves computing importance scores of each feature dimension in $\mathbf{x}$ on the prediction of a given instance. Examples for methods that provide this type of explanation include SHAP [1], INVASE [6], DeepLIFT [8], L2X [15], LIME [10, 16], GAM [37], and Saliency maps [39]. Each of these methods follows one of two approaches. The first approach, adopted by saliency maps, use the *gradients* of the model output with respect to the input as a measure of feature importance. The second approach, followed by LIME, DeepLIFT, GAM and SHAP, uses *local additive approximations* to explicitly quantify the additive contribution of each feature.

Symbolic metamodeling enables a *unified* framework for (instancewise) feature importance scoring that encapsulates the two main approaches in the literature. To show how this is possible, consider the following Taylor expansion of the metamodel $g(\mathbf{x})$ around a feature point $\mathbf{x}_0$:

$$g(\mathbf{x}) = g(\mathbf{x}_0) + (\mathbf{x} - \mathbf{x}_0) \cdot \nabla_\mathbf{x} g(\mathbf{x}_0) + (\mathbf{x} - \mathbf{x}_0) \cdot \boldsymbol{H}(\mathbf{x}) \cdot (\mathbf{x} - \mathbf{x}_0) + \ldots, \quad (7)$$

where $\boldsymbol{H}(\mathbf{x}) = [\partial^2 g / \partial x_i \partial x_j]_{i,j}$ is the Hessian matrix. Now consider — for simplicity of exposition — a second-order approximation of (7) with a two-dimensional feature space $\mathbf{x} = (x_1, x_2)$, i.e.,

$$
\begin{aligned}
g(\mathbf{x}) \approx\ & g(\mathbf{x}_0) + (x_1 - x_{0,1}) \cdot g_{x_1}(\mathbf{x}_0) - x_{0,2} \cdot x_1 \cdot g_{x_1 x_2}(\mathbf{x}_0) + \tfrac{1}{2}(x_1 - x_{0,1})^2 g_{x_1 x_1}(\mathbf{x}_0) \\
& + (x_2 - x_{0,2}) \cdot g_{x_2}(\mathbf{x}_0) - x_{0,1} \cdot x_2 \cdot g_{x_1 x_2}(\mathbf{x}_0) + \tfrac{1}{2}(x_2 - x_{0,2})^2 g_{x_2 x_2}(\mathbf{x}_0) \\
& + x_1 \cdot x_2 \cdot g_{x_1 x_2}(\mathbf{x}_0),
\end{aligned}
\quad (8)
$$

where $g_x = \nabla_x g$ and $\mathbf{x}_0 = (x_{0,1}, x_{0,2})$. In (8), the term in blue (first line) reflects the importance of feature $x_1$, the term in red (second line) reflects the importance of feature $x_2$, whereas the last term (third line) is the interaction between the two features. The first two terms are what generalized additive models, such as GAM and SHAP, compute. LIME is a special case of (8) that corresponds

to a first-order Taylor approximation. Similar to saliency methods, the feature contributions in (8) are computed using the gradients of the model with respect to the input, but (8) is more general as it involves higher order gradients to capture the feature contributions more accurately. All the gradients in (8) can be computed efficiently since the exact gradient of the $G$ function with respect to its input can be represented analytically in terms of another $G$ function (see Appendix A).

Statistical interactions between features are another form of model interpretation that has been recently addressed in [9, 13]. As we have seen in (8), feature interactions can be analytically derived from a symbolic metamodel. The series in (8) resembles the structure of the pairwise interaction model GA$^2$M in [9] and the NIT disentanglement method in [13]. Unlike both methods, a symbolic metamodel can analytically quantify the strength of higher-order (beyond pairwise) interactions with no extra algorithmic complexity. Moreover, unlike the NIT model in [13], which is tailored to neural networks, a symbolic metamodel can quantify the interactions in any machine learning model (7).

Table 2: Comparison between SM and SR.

|  | $f_1(x) = e^{-3x}$ | $f_2(x) = \frac{x}{(x+1)^2}$ | $f_3(x) = \sin(x)$ | $f_4(x) = J_0(10\sqrt{x})$ |
|---|---|---|---|---|
| **SM$^p$** | $-x^3 + \frac{5}{2}(x^2 - x) + 1$ | $\frac{x^3}{3} - \frac{4x^2}{5} + \frac{2x}{3}$ | $\frac{-1}{4}x^2 + x$ | $-7(x^2 - x) - 1.4$ |
|  | $R^2$: 0.995 | $R^2$: 0.985 | $R^2 : 0.999$ | $R^2 : -4.75$ |
| **SM$^c$** | $x^{4 \times 10^{-6}} e^{-2.99x}$ | $x(x+1)^{-2}$ | $1.4\, x^{1.12}$ | $I_{0.0003}\left(10\, e^{\frac{j\pi}{2}}\sqrt{x}\right)$ |
|  | $R^2$: 0.999 | $R^2$: 0.999 | $R^2 : 0.999$ | $R^2 : 0.999$ |
| **SR** | $x^2 - 1.9x + 0.9$ | $\frac{0.7x}{x^2 + 0.9x + 0.75}$ | $-0.17\, x^2 + x + 0.016$ | $-x(x - 0.773)$ |
|  | $R^2$: 0.970 | $R^2$ 0.981 | $R^2 : 0.998$ | $R^2 : 0.116$ |

## 5 Experiments and Use Cases

Building on the discussions in Section 4, we demonstrate the use cases of symbolic metamodeling through experiments on synthetic and real data. In all experiments, we used `Sympy` [19] (a symbolic computation library in Python) to carry out computations involving Meijer $G$-functions[5].

### 5.1 Synthetic experiments

**Can we learn complex symbolic expressions?** We start off with four synthetic experiments with the aim of evaluating the richness of symbolic expressions discovered by our metamodeling algorithm. In each experiment, we apply Algorithm 1 (Section 3.2) on a ground-truth univariate function $f(x)$ to fit a metamodel $g(x) \approx f(x)$, and compare the resulting mathematical expression for $g(x)$ with that obtained by Symbolic regression [25], which we implement using the `gplearn` library [40].

In Table 2, we compare symbolic metamodeling (SM) and symbolic regression (SR) in terms of the expressions they discover and their $R^2$ coefficient with respect to the true functions. We consider four functions: an exponential $e^{-3x}$, a rational $x/(x+1^2)$, a sinusoid $\sin(x)$ and a *Bessel function* of the first kind $J_0(10\sqrt{x})$. We consider two versions of SM: SM$^p$ for which $\mathcal{G} =$ Polynomial expressions, and SM$^c$ for which $\mathcal{G} =$ Closed-form expressions. As we can see, SM is generally more accurate and more expressive than SR. For $f_1(x)$, $f_2(x)$ and $f_4(x)$, SM managed to figure out the functional forms of the true functions ($J_0(x) = I_0(e^{\frac{j\pi}{2}} x)$), where $I_0(x)$ is the *Bessel function of the second kind*. For $f_3(x)$, SM$^c$ recovered a parsimonious approximation $g_3(x)$ since $\sin(x) \approx x$ for $x \in [0, 1]$. Moreover, SM$^p$ managed to retrieve more accurate polynomial expressions than SR.

**Instancewise feature importance.** Now we evaluate the ability of symbolic metamodels to explain predictions in terms of instancewise feature importance (Section 4). To this end, we replicate the experiments in [15] with the following synthetic data sets: *XOR*, *Nonlinear additive features*, and *Feature switching*. (See Section 4.1 in [15] or Appendix B for a detailed description of the data sets.) Each data set has a 10-dimensional feature space and 1000 data samples.

For each of the three data sets above, we fit a 2-layer neural network $f(\mathbf{x})$ (with 200 hidden units) to predict the labels based on the 10 features, and then fit a symbolic metamodel $g(\mathbf{x})$ for the trained

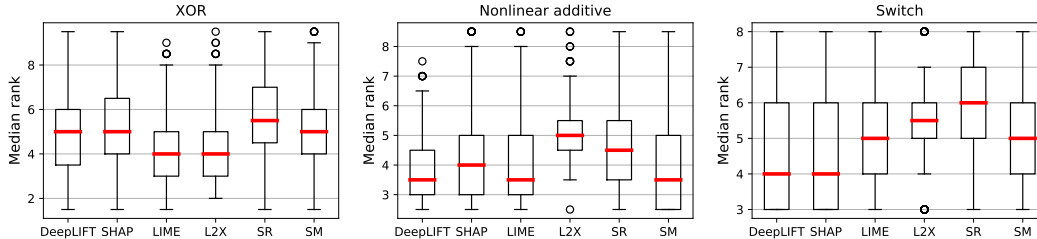

Figure 4: Box-plots for the median ranks of features by their estimated importance per sample over the 1000 samples of each data set. The red line is the median. Lower median ranks are better.

network $f(\mathbf{x})$ using the algorithm in Section 3.2. Instancewise feature importance is derived using the (first-order) Taylor approximation in (8). Since the underlying true features are known for each sample, we use the median feature importance ranking of each algorithm as a measure of the accuracy of its feature ranks as in [15]. Lower median ranks correspond to more accurate algorithms.

In Figure 4, we compare the performance of metamodeling (SM) with DeepLIFT, SHAP, LIME, and L2X. We also use the Taylor approximation in (8) to derive feature importance scores from a symbolic regression (SR) model as an additional benchmark. For all data sets, SM performs competitively compared to L2X, which is optimized specifically to estimate instancewise feature importance. Unlike LIME and SHAP, SM captures the strengths of feature interactions, and consequently it provides more modes of explanation even in the instances where it does not outperform the additive methods in terms of feature ranking. Moreover, because SM recovers more accurate symbolic expressions than SR, it provides a more accurate feature ranking as a result.

## 5.2 Predicting prognosis for breast cancer

We demonstrate the utility of symbolic metamodeling in a real-world setup for which model interpretability and *transparency* are of immense importance. In particular, we consider the problem of predicting the risk of mortality for breast cancer patients based on clinical features. For this setup, the *ACCJ* guidelines require prognostic models to be formulated as transparent equations [21] — symbolic metamodeling can enable machine learning models to meet these requirements by converting black-box prognostic models into risk equations that can be written on a piece of paper.

Using data for 2,000 breast cancer patients extracted from the UK cancer registry (data description is in Appendix B), we fit an XGBoost model $f(\mathbf{x})$ to predict the patients' 5 year mortality risk based on 5 features: age, number of nodes, tumor size, tumor grade and Estrogen-receptor (ER) status. Using 5-fold cross-validation, we compare the area under receiver operating characteristic (AUC-ROC) accuracy of the XGBoost model with that of the PREDICT risk calculator (https://breast.predict.nhs.uk/), which is the risk equation most commonly used in current practice [41]. The results in Table 3 show that the XGBoost model provides a statistically significant improvement over the PREDICT score.

Using our metamodeling algorithm (with $\mathcal{G}$ set to be the space of closed-form expressions), we obtained the symbolic metamodel for both the XGBoost and PREDICT models. As we can see in Figure 5, by inspecting the median instancewise feature ranks, we can see that PREDICT overestimates the importance of some features and underestimates that of others. This gives us an indication as to why XGBoost was able to achieve a gain in predictive accuracy.

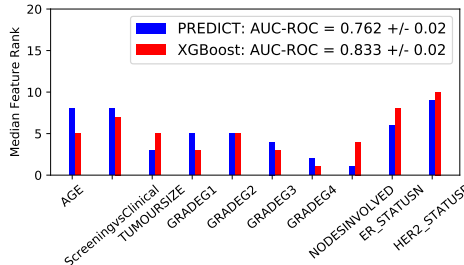

Figure 5: Feature importance for PREDICT and XGBoost.

Through a symbolic equation, clinicians can transparently use the accurate prognostic model learned by XGBoost, without worrying about its original black-box nature. The transparent nature of the metamodel not only ensures its trustworthiness, but also helps us understand the sources of performance gain achieved by the original XGBoost model. Moreover, using the metamodel, we were able to draw insights into the impact of the interactions between ER, number of nodes and tumor size on a patient's risk. Such insights would be very hard to distill from the original XGBoost model.

## Acknowledgments

This work was supported by the National Science Foundation (NSF grants 1462245 and 1533983), and the US Office of Naval Research (ONR). The data for our experiments was provided by the UK national cancer registration and analysis service (NCRAS).

## Footnotes

[1]Note that here we are concerned with explaining the predictions of a trained model, i.e., its *response surface*. Other works, such as [20], focus on explaining the model's *loss surface* in order to understand how it learns.

[2]For instance, for an $L$-layer neural network, $\mathcal{F}$ is the space of compositions of $L$ nested activation functions. For a random forest with $L$ trees, $\mathcal{F}$ is the space of summations of $L$ piece-wise functions.

[3]The Kolmogorov decomposition in (3) is a universal function approximator [31]. In fact, (3) can be thought of as a 2-layer neural network with generalized activation functions [32–34, 31, 35].

[4]Since $\Gamma(x) = (x-1)!$, the zeros of factors $\Gamma(b_j - s)$ and $\Gamma(1 - a_j + s)$ are $(b_j - k)$ and $(1 - a_j - k)$, $k \in \mathbb{N}_0$, respectively, whereas the poles of $\Gamma(1 - b_j + s)$ and $\Gamma(a_j + s)$ are $(-a_j - k)$ and $(1 - b_j - k)$, $k \in \mathbb{N}_0$.

[5]The code is provided at https://bitbucket.org/mvdschaar/mlforhealthlabpub.

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
