[Supplementary Material]

# Supplementary Material for "Demystifying Black-box Models with Symbolic Metamodels"

**Ahmed M. Alaa**
ECE Department
UCLA
ahmedmalaa@ucla.edu

**Mihaela van der Schaar**
University of Cambridge, UCLA, and
Alan Turing Institute
{mv472@cam.ac.uk,mihaela@ee.ucla.edu}

## Appendix A: Meijer-G Functions

The Meijer $G$-functions are a family of univariate functions, each of them determined by finitely many indexes. The $G$-function is a linear combination of certain special functions of standard type, and hence it is often use to solve indefinite integrals [1]. Despite their useful properties, $G$-functions are not very well known in the mathematical community. In this Section, we provide a brief background on Meijer $G$-functions, presenting its formal definition along with its basic properties.

### Definition

The Meijer-G Function is a univariate function given by a line integral in the complex plane [2, 3]:

$$G_{p,q}^{m,n}\left(\begin{smallmatrix} a_1,\ldots,a_p \\ b_1,\ldots,b_q \end{smallmatrix} \Big| x\right) = \frac{1}{2\pi i} \int_{\mathcal{L}} \frac{\prod_{j=1}^{m} \Gamma(b_j - s) \prod_{j=1}^{n} \Gamma(1 - a_j + s)}{\prod_{j=m+1}^{q} \Gamma(1 - b_j + s) \prod_{j=n+1}^{p} \Gamma(a_j + s)}\, x^s\, ds, \qquad (0.1)$$

where $\Gamma(.)$ is the Gamma function, $0 \leq m \leq q$, $0 \leq n \leq p$, where $m$, $n$, $p$ and $q$ are integer numbers, $a_k - b_j \neq 1, 2, 3, \ldots$ for $k = 1, 2, \ldots, n$, $j = 1, 2 \ldots, m$ and $x \neq 0$. This means that no pole of any $\Gamma(b_j - s)$ coincides with any pole of any $\Gamma(1 - a_k + s)$. In (0.1), the integration path $L$ separates the poles of the factors $\Gamma(b_j - s)$ from those of the factors $\Gamma(1 - a_k + s)$.

**Case 1**        **Case 2**        **Case 3**

The integration path $L$ in (0.1) can be chosen as follows [3, 4, 1]:

1. $L$ goes from $-i\infty$ to $i\infty$.

2. $L$ is a loop that starts at $i\infty$ on a line parallel to the $+$ve real axis, encircles the poles of the factors $\Gamma(b_j - s)$ and returns to $i\infty$ on another line parallel to the $+$ve real axis.

3. $L$ is a loop that starts at $i\infty$ on a line parallel to the $-$ve real axis, encircles the poles of the factors $\Gamma(1 - a_k + s)$ and returns to $\infty$ on another line parallel to the $-$ve real axis.

The integration paths for the three cases are illustrated on the complex plane in the Figure above. When more than one of Cases 1, 2, and 3 is applicable the same value is obtained for the $G$-function

The conditions for convergence of the line integral can be established by applying Stirling's asymptotic approximation to the gamma functions $\Gamma(b_j - s)$ and $\Gamma(1 - a_k + s)$ in the integrand. As a consequence of the definition above, the Meijer $G$-function is an analytic function of $x$ (with possible exception of the origin $x = 0$ and the unit circle $|x| = 1$).

Another definition of the $G$-function is as a solution to a linear differential equations. In particular, the $G$-function satisfies the following linear differential equation of order $\max(p, q)$:

$$\left[ (-1)^{p-m-n} \, x \prod_{j=1}^{p} \left( x \frac{d}{dx} - a_j + 1 \right) - \prod_{j=1}^{q} \left( x \frac{d}{dx} - b_j \right) \right] G(x) = 0. \tag{0.2}$$

In the rest of this Section, we present the basic properties of the $G$-function in addition to some of its special values for which the $G$-function reduces to familiar special functions.

**Basic properties**

*Order reduction*

From the definition of the $G$-function, we can see that if equal parameters appear among the $(a_1, \ldots, a_p)$ and $(b_1, \ldots, b_p)$ factors in the numerator and the denominator of the integrand, then some poles and zeros would cancel each other and the integrand will be simplified, i.e., the order $m$ or $n$ will be reduced. That is, we have the following:

$$G_{p,q}^{m,n}\left( \begin{array}{c} a_1, a_2, \ldots, a_p \\ b_1, \ldots, b_{q-1}, a_1 \end{array} \,\middle|\, x \right) = G_{p-1,\, q-1}^{m,\, n-1}\left( \begin{array}{c} a_2, \ldots, a_p \\ b_1, \ldots, b_{q-1} \end{array} \,\middle|\, x \right), \quad n, p, q \geq 1$$

$$G_{p,q}^{m,n}\left( \begin{array}{c} a_1, \ldots, a_{p-1}, b_1 \\ b_1, b_2, \ldots, b_q \end{array} \,\middle|\, x \right) = G_{p-1,\, q-1}^{m-1,\, n}\left( \begin{array}{c} a_1, \ldots, a_{p-1} \\ b_2, \ldots, b_q \end{array} \,\middle|\, x \right), \quad m, p, q \geq 1. \qquad (0.3)$$

This property is important because it means that we can have an optimization domain with fixed dimensions in the Bayesian optimization algorithm. In order to span arbitrary number of parameters $p$ and $q$, we can set some of the parameters in $(a_1, \ldots, a_p)$ and $(b_1, \ldots, b_p)$ to the same values so that some of the Gamma factors in the denominator and the numerator cancel each other.

*G-function identities*

To gain insight into the flexibility of the $G$-functions, we present the following identities [2, 3]:

$$x^\mu \, G_{p,q}^{m,n}\left( \begin{array}{c} a_1, \ldots, a_p \\ b_1, \ldots, b_p \end{array} \,\middle|\, x \right) = G_{p,q}^{m,n}\left( \begin{array}{c} a_1 + \mu, \ldots, a_p + \mu \\ b_1 + \mu, \ldots, b_p + \mu \end{array} \,\middle|\, x \right),$$

$$G_{p,q}^{m,n}\left( \begin{array}{c} a_1, \ldots, a_p \\ b_1, \ldots, b_p \end{array} \,\middle|\, x \right) = G_{q,p}^{n,m}\left( \begin{array}{c} 1 - b_1, \ldots, 1 - b_p \\ 1 - a_1, \ldots, 1 - a_p \end{array} \,\middle|\, x^{-1} \right). \qquad (0.4)$$

This means that if a $G$-function correspond to an exponential function $\exp(x)$, then we can recover the function $\exp(1/x)$ by using the parameters $(1 - a_1, \ldots, 1 - a_q)$ and $(1 - b_1, \ldots, 1 - b_q)$ instead of $(a_1, \ldots, a_p)$ and $(b_1, \ldots, b_q)$. Similarly, we can recover a function $x^2 \exp(x)$ by adding 2 to the parameters $(a_1, \ldots, a_p)$ and $(b_1, \ldots, b_q)$.

*Derivatives*

The $k^{th}$ derivative of the $G$-function with respect to the input $x$ is given by [1]:

$$x^h \frac{d^h}{dx^h} \, G_{p,q}^{m,n}\left( \begin{array}{c} a_1, \ldots, a_p \\ b_1, \ldots, b_p \end{array} \,\middle|\, x \right) = G_{p+1,\, q+1}^{m,\, n+1}\left( \begin{array}{c} 0, a_1, \ldots, a_p \\ b_1, \ldots, b_p, h \end{array} \,\middle|\, x \right)$$

$$= (-1)^h \, G_{p+1,\, q+1}^{m+1,\, n}\left( \begin{array}{c} a_1, \ldots, a_p, 0 \\ h, b_1, \ldots, b_p \end{array} \,\middle|\, x \right). \qquad (0.5)$$

Thus, we can always obtain the derivatives of a function represented in terms of a $G$-function. The derivative is also a $G$-function, and hence $G$-functions are closed under differentiation.

The relation above is satisfied for $h < 0$ as well, which means that the antiderivative of a $G$-function can be computed as easily as the derivative, and can always be represented as a $G$-function as well.

**Representation of familiar functions in terms of the $G$-function**

The main advantage of modeling univariate functions using a $G$-function representation is that the $G$-function reduces to almost all familiar functional forms (closed-form, algebraic and analytic expressions, in addition to special functions) as special cases. Thus, a $G$-function gives us a parameterizable space of diverse functional forms of interest.

The following table lists some special cases where the $G$-function reduces to familiar functional forms for different settings of the parameters $(a_1, \ldots, a_p, b_1, \ldots, b_q)$.

| $G$-function | Equivalent function | $G$-function | Equivalent function |
|:---:|:---:|:---:|:---:|
| $G_{0,1}^{1,0}\left(\begin{smallmatrix}-\\0\end{smallmatrix}\middle\vert -x\right)$ | $e^x$ | $G_{2,2}^{1,2}\left(\begin{smallmatrix}\frac{1}{2},1\\\frac{1}{2},0\end{smallmatrix}\middle\vert x^2\right)$ | $2\arctan(x)$ |
| $G_{2,2}^{1,2}\left(\begin{smallmatrix}1,1\\1,0\end{smallmatrix}\middle\vert x\right)$ | $\log(1+x)$ | $G_{1,2}^{2,0}\left(\begin{smallmatrix}1\\\alpha,0\end{smallmatrix}\middle\vert x\right)$ | $\Gamma(\alpha,x)$ |
| $G_{0,2}^{1,0}\left(\begin{smallmatrix}-\\0,\frac{1}{2}\end{smallmatrix}\middle\vert \frac{x^2}{4}\right)$ | $\frac{1}{\sqrt{\pi}}\cos(x)$ | $G_{1,2}^{2,0}\left(\begin{smallmatrix}1\\0,\frac{1}{2}\end{smallmatrix}\middle\vert x^2\right)$ | $\sqrt{\pi}\,\mathrm{erfc}(x)$ |
| $G_{0,2}^{1,0}\left(\begin{smallmatrix}-\\\frac{1}{2},0\end{smallmatrix}\middle\vert \frac{x^2}{4}\right)$ | $\frac{1}{\sqrt{\pi}}\sin(x)$ | $G_{0,2}^{1,0}\left(\begin{smallmatrix}-\\\frac{a}{2},\frac{-a}{2}\end{smallmatrix}\middle\vert \frac{x^2}{4}\right)$ | $J_a(x)$ |

To illustrate the expressive power of $G$-functions, consider the following example of a $G$-function parametrized by two parameters $a$ and $b$ as follows:

$$\widehat{f}(x;a,b) = G_{2,2}^{1,2}\left(\begin{smallmatrix}a,a\\a,b\end{smallmatrix}\middle\vert x\right).$$

As we can see in this figure, the function $\widehat{f}(x;a,b)$ takes different forms for different settings of the parameters $a$ and $b$. For instance, $\widehat{f}(x;a=1,b=1)$ is a rational function $x/(x+1)$, $\widehat{f}(x;a=1,b=0)$ is a logarithmic function $\log(x+1)$, whereas $\widehat{f}(x;a=0,b=-1)$ is $\frac{1}{x}\log(x+1)$. Thus, by tuning only 2 parameters, $G_{2,2}^{1,2}\left(\begin{smallmatrix}a,a\\a,b\end{smallmatrix}\middle\vert x\right)$ reduces to different functional forms. These forms are only limited to rational and logarithmic functions only because we restricted the parametrization of the $G$-function to $G_{2,2}^{1,2}\left(\begin{smallmatrix}a,a\\a,b\end{smallmatrix}\middle\vert x\right)$. To model more diverse forms for $f$, we can use a richer parametrization to recover the functions in the Table above.

## Appendix B: Experimental Details

**Synthetic data sets**

we replicate the experimental setup in [5] with the following benchmark synthetic datasets:

- **2-dimensional XOR as binary classification.** The feature $\mathbf{x}$ is a 10-dimensional standard Gaussian. The response is generated from $\mathbb{P}(y=1\,|\,\mathbf{x}) \propto \exp\{x_1 x_2\}$.
- **Orange skin.** The feature $\mathbf{x}$ is a 10-dimensional standard Gaussian. The response is generated from $\mathbb{P}(y=1\,|\,\mathbf{x}) \propto \exp\{x_1 + x_2 + x_3 + x_4 - 4\}$.
- **Nonlinear additive model.** The feature $\mathbf{x}$ is generated from a 10-dimensional standard Gaussian, and $\mathbb{P}(y=1\,|\,\mathbf{x}) \propto \exp\{-100\sin(2x_1) + 2|x_2| + x_3 + \exp\{-x_4\}\}$.
- **Switch feature.** Generate $x_1$ from a mixture of two Gaussians centered at $\pm3$ respectively with equal probability. If $x_1$ is generated from the Gaussian centered at 3, the 2-5th dimensions are used to generate $y$ like the orange skin model. Otherwise, the 6-9th dimensions are used to generate $y$ from the nonlinear additive model.

The first three datasets are commonly used datasets in the feature selection literature [6]. The first dataset is designed specifically for feature interaction, whereas the fourth dataset is designed for instancewise feature selection. In all these datasets, ground-truth about functional forms, feature importance and interactions are known.

**Breast cancer data set**

The UK Cancer Database is a registry of patients with diagnosed breast cancer from 2000 onwards, with 601,272 patients having been diagnosed between the years 2000 and 2016, with a median follow

up of six years. The database includes women between 30-90 years of age. (We only included women >35 years of age). We excluded patients who did not undergo surgery, as these were not included in the original study for which the competing PREDICT score was fit. We restricted our training and internal validation set to individuals who had information on all variables used by PREDICT. Our final training data set included 50,288 individuals, with a median follow up of 6 years.