[Reviews · NeurIPS 2019]

Reviewer 1



Originality: although similar in spirit to past works on approximating black-box models with interpretable surrogates / performing symbolic regression (which it does a good job citing), this work seems to go several steps further and contribute something novel both in framing the problem and solving it. (It doesn't seem like past work on symbolic regression use Meijer G-functions.) Quality: the work seems technically sound, though it makes a number of arbitrary choices and I think it could do with more experiments. I'll elaborate on these points in the improvements section. Clarity: the paper is very clearly written and its results are nicely presented. Significance: I think this paper has the potential to become very significant, with immediate practical applications in medicine and accelerated science and many areas for follow-up research.

Reviewer 2



I am new to the domain of symbolic regression and found the article to constitute a well-written and interesting introduction to it. Yet, I kept wondering to what extent the presented approach can really help interpreting complex black box functions. In the final example, it is clear that the results are fairly simple and interpretable while delivering a moderate loss in prectivity compared to the crude algorithm. But in more generality, I still don't see how combinations of Bessel functions and alike will help most practitioners. Which leads us to a question that to the best of my understanding was somehow underinvestigated here, namely some more systematic approach on how to tune the complexity of the metamodel, and maybe explore the Pareto front of simplicity versus predictivity. Besides this, I keep also wondering why such approach should be restricted to functions returned by ML algorithm and could not be trained directly based on raw data? On a different question, here an L2 loss is considered, and furthermore the (meta)model fitting is performed by (local) gradient descent. Why not consider on the one hand some more general class of misfits, and on the other hand allow more varied classes of (global) optimization algorithms? Unless I have overlooked an important point, I don't see why the fitting problem at hand should be convex? Last but not least, I was surprised not to find more references/comparisons to further machine learning approaches magnifying interpretability, e.g. those based on kernels and their decompositions, with a variety of methods encompassing Additive/ANOVA GPs and splines (Durrande, Duvenaud, Plate, Wahba, etc.), High Dimensional Model Representation, etc: T.A. Plate. Accuracy versus interpretability in flexible modeling: Implementing a tradeoff using Gaussian process models. Behaviormetrika, 26:29–50, 1999. Li et al. Global uncertainty assessments by high dimensional model representations (HDMR). Chemical Engineering Science. Volume 57, Issue 21, Pages 4445-4460 (2002). Whaba, G. et al. Smoothing Spline Anova for Exponential Families, with Application to the Wisconsin Epidemiological Study of Diabetic Retinopathy. The Annals of Statistics Vol. 23, No. 6, pp. 1865-1895 (1995) Duvenaud, D. Nickisch, H. and Rasmussen, C.E. Additive Gaussian Processes. Neural Information Processing Systems (2011) Durrande, N., Ginsbourger, D. and Roustant, O. Additive covariance kernels for high-dimensional Gaussian process modeling. Annales de la Faculté des sciences de Toulouse: Mathématiques 21 (3), 481-499 (2012). Duvenaud, D. et al. Structure Discovery in Nonparametric Regression through Compositional Kernel Search. ICML 2013. Durrande, N. et al. ANOVA kernels and RKHS of zero mean functions for model-based sensitivity analysis Journal of Multivariate Analysis 115, 57-67 (2013). Finally, the term "metamodelling" is also vastly used in the domain of "Computer Experiments", see works of authors including Santner, Wynn, Schonlau, etc. See for instance Santer, T.J., Williams, B.J and Notz, W.I. The Design and Analysis of Computer Experiments. Springer 2003 and references therein such as Sacks et al. Design and Analysis of Computer Experiments. Statistical Science. Vol. 4, No. 4, pp. 409-423 (1989) and Jones, D.R., Schonlau, M. & Welch, W.J. Journal of Global Optimization (1998) 13: 455. https://doi.org/10.1023/A:1008306431147 ******** Update afer rebuttal ********** I found the rebuttal quite constructive and while I did not get completely rid of formerly expressed reservations (regarding the actual interpretability of classes of functions appealed to as well as some arbitrariness in the way regularization is performed), I feel that the work has improved and that it is worth investigating further such approaches and potential practical benefits. As a consequence, I increased my score by one unit.

Reviewer 3



After reading reviews and author feedback I raise my score to 7. The authors responded to my concern and I think this submission is a good contribution. ----------- **tl;dr** The ability to learn interpretable meta-models with gradient descent is a good contribution. Some more empirical evidence would make the paper’s claim stronger. **Summary** The authors propose an elegant way to create an interpretable, symbolic model via regressing the target model's function. By using Meijer G-function they are able to learn the structure of the symbolic model via gradient descent, in contrast to prevailing genetic methods. Building on the constructed meta-model the paper shows how these models can be seen as a unification of two branches of the interpretation literature: feature importance estimators and local additive approximations. Finally an empirical evaluation on synthetic data and a real world dataset is provided. The use case on cancer data is very interesting. **Originality** The original contribution of this paper is to show a way to learn symbolic models via gradient descent rather than with genetic methods. **Quality** The overall quality of the paper is good. Yet a larger/better empirical evaluation is needed: * When does this approach not work anymore? E.g., how about input with a large feature dimensions? models that create rich internal feature representations? * Which results are reached by competitors on the given example on breast cancer? Is there a runtime benefit compared to genetic models? * Cancer data result: Did you reach to this result at every experiment run? What is the variance due to gradient descent of your approach? * What is the influence of the hyper-parameters m, n, p, q, r? **Clarity** The paper is well written and easy to read. **Significance** The significance is medium. The original contribution is there, but it seems the application is limited (or at least it was not shown that the method works on more complex models/input data for which such a method is needed most). **Questions** * Genetic models can build trees of symbolic expressions, your approach seems to be limited to additive meta-models. Is this right? Is this a limitation? * SHAPE also unifies the same branches of the interpretation literature, can you please set this into context?

[Author Response · NeurIPS 2019]

[[ **Reviewer 1** ]] Thank you for your feedback. We will definitely release our code along with the camera-ready version of the manuscript. ∎ **Fitting to training data:** The advantage of fitting the meta-model on sampled feature points is that the accuracy of the meta-model would not be limited by the size of the training data. However, if the meta-model is meant to be optimized w.r.t the feature distribution, then one can fit the feature distribution, say using a GAN or a kernel density function, and sample feature points from the estimated distribution to train the meta-model. Fitting the meta-model directly on training data will correspond to a 2-layer neural network with Meijer-$G$ function as activation functions (see Figure 3). While this is very interesting, it departs from the main objective of the paper and demands a separate analysis on generalization performance, so we will add this discussion in the supplementary material. ∎ **Loss function & regularization:** The loss function should be selected based on the application, e.g., if $f$ is a classifier, then $\ell$ should be a cross-entropy loss. The idea of adding a regularization term is also very interesting although it is not straightforward. We will investigate using the number of poles and zeros as a penalty term as it is a natural measure of the complexity of a $G$ function. We will add a discussion on loss functions and regularization in the final manuscript.

[[ **Reviewer 2** ]] Thank you for your helpful comments and suggestions. ∎ **Interpretability of complicated functions:** As mentioned in lines 75 and 89, different functional forms are deemed interpretable in different applications. Bessel functions (and other special functions) are very common in empirical physics and material sciences (e.g. wave and field equations are modeled with such functions [3, 4]). (Please also refer to response Significance & applicability for Reviewer 3.) The theoretical justification of our framework was provided in Section 3.1, where we have shown that — based on the Kolomogorov superposition theorem — our approach can approximate any multivariate continuous function. ∎ **Complexity tuning:** Our algorithm explores the Pareto front of simplicity vs. predictivity systematically in two ways: (1) it uses Bayesian optimization to conduct hyper-parameter search by picking the smallest number of poles and zeros for the Meijer-$G$ function (i.e., simplest functional form) that best fits the model, and (2) it uses polynomial Chebyshev approximations to simplify meta-models with complex functional forms (Algorithm 1). We will emphasize this in the final manuscript. ∎ **Fitting to training data:** Please kindly refer to response Fitting to training data for Reviewer 1. ∎ **Loss function:** Our framework does not pose limitations on the loss function being used: any differentiable loss function (e.g., cross-entropy) can be used instead of the $L$-2 loss in Equation (2). ∎ **Convexity:** In general, optimizing symbolic models with arbitrary non-linearity cannot be formulated as a convex optimization problem unless strict prior assumptions on the symbolic functions (e.g., linearity) are made (as in [8, 14]). This is why symbolic regression models resort to search algorithms based on genetic programming, which also does not guarantee a global solution [23-25]. Moreover, most of the competitive deep learning-based baselines such as DeepLIFT and L2X also use gradient descent. A key strength of our framework is that for the first time, flexible symbolic modeling can be conducted efficiently via gradient descent rather than exhaustive search heuristics. We believe this to be a strength of our method and not a weakness. ∎ **Extra references:** We will add all the suggested references in the final the manuscript.

In addition, we have implemented two of the requested baselines and incorporated the results into Sections 5.1 and 5.2. The two baselines are: the additive GP by Duvenaud et al. and ANOVA GP by Kaufman et al.. As shown in the following Table, we found that neither baselines outperformed our model for experiment 5.2. Our interpretation for these results is that the additive GP kernel decomposition cannot capture the intricate interactions between (overlapping) feature subsets learned by the reference XGBoost model.

|  | AUC-ROC |
| --- | --- |
| **SM** | $0.8651 \pm 0.0045$ |
| **Additive GP** | $0.8502 \pm 0.0062$ |
| **ANOVA GP** | $0.8498 \pm 0.0053$ |

[[ **Reviewer 3** ]] Thank you for your valuable comments. ∎ **Significance & applicability:** As mentioned in lines 66-76 and Section 4, our method is applicable to the wide range of setups where a model's feature importance, interactions or explicit equations are essential for understanding its instance-wise predictions or uncovering the sources of its performance gain. We demonstrated the significance of our algorithm through the exemplary medical application in Section 5.2, which entailed explaining the predictions of a complex model for breast cancer, and helped recover new feature interactions that were unknown in the clinical literature. We will make sure that these aspects regarding the significance of our work are clearly stated in the camera-ready version of the paper. ∎ **Empirical evaluation:** By virtue of the Kolomogorov superposition theorem [28], our algorithm can model any multivariate continuous function regardless of its dimensionality and the richness of its internal feature representations. Our algorithm is in fact more advantageous for more complex models since gradient descent is more efficient in large parameter spaces compared to black-box optimization methods which scale exponentially with the number of parameters. In the final manuscript, we will add the AUC-ROC performance of symbolic regression (SR) to Table 3. The run-time of SR on this dataset was 3.5 times longer than our algorithm. The functional form of the equation in line 267 was the same in all 5 runs, and the variability of the coefficients across runs was statistically insignificant. We will report the variance of the coefficients in line 267 in the supplementary material. ∎ **Influence of hyper-parameters:** More complex models require more poles and zeros (hyper-parameters) for the corresponding meta-model. We tuned the hyper-parameters in Section 5.2 using Bayesian optimization. ∎ **Related literature:** In the final version of the paper, we will make it clear that our framework does not encompass the line of research including LRP, PatternAttribution/Net, DeepTaylor, etc, and will point out to the unifying nature of the SHAP framework. ∎ **Limits on symbolic expressions:** Our approach is not limited to additive meta-models: as can be seen in equation (5), our meta-models comprise composite (nested) functions of additive functions of the form $\sum_j f_j(g_1^j(x_1) + \ldots + g_n^j(x_n))$. By expanding these composite functions (e.g., using Taylor's expansion) we can recover rich multiplicative terms similar to those in the expression trees of genetic models.

[Meta-Review · NeurIPS 2019]

The reviewers agreed that this paper presents a valuable contribution; they appreciated the quality of the writing, the overall motivation of interpretability of the models, and the proposed approach to use gradient descent to learn symbolic models. The primary shortcomings that the discussion focused around were some of the limited evaluation included in the paper, and the justification of the proposed models as being “interpretable”. The response was useful in addressing some of these concerns.